# Accuracy of a Smartwatch to Assess Heart Rate Monitoring and Atrial Fibrillation in Stroke Patients

**DOI:** 10.3390/s23104632

**Published:** 2023-05-10

**Authors:** Claudia Meza, Jesus Juega, Jaume Francisco, Alba Santos, Laura Duran, Maite Rodriguez, Jose Alvarez-Sabin, Laia Sero, Xavier Ustrell, Saima Bashir, Joaquín Serena, Yolanda Silva, Carlos Molina, Jorge Pagola

**Affiliations:** 1Stroke Unit, Department of Neurology, Hospital Universitari Vall d’Hebron, 08035 Barcelona, Spain; 2Vall d’Hebron Institut de Recerca (VHIR), 08035 Barcelona, Spain; 3Department of Medicine, Faculty of Medicine, Hospital Vall d’Hebron, Universitat Autonoma de Barcelona, 08193 Barcelona, Spain; 4Arrhythmia Unit, Department of Cardiology, Hospital Universitari Vall d’Hebron, 08035 Barcelona, Spain; 5Department of Neurology, Hospital Universitari Joan XXIII, 43005 Tarragona, Spain; 6Cerebrovascular Pathology Research Group, Girona Biomedical Research Institute (IDIBGI), 17790 Girona, Spainysilva.girona.ics@gencat.cat (Y.S.); 7Department of Neurology, Hospital Universitari de Girona Dr. JosepTrueta, 17007 Girona, Spain

**Keywords:** atrial fibrillation, stroke, wearable, smartwatch, screening, accuracy

## Abstract

(1) Background: Consumer smartwatches may be a helpful tool to screen for atrial fibrillation (AF). However, validation studies on older stroke patients remain scarce. The aim of this pilot study from RCT NCT05565781 was to validate the resting heart rate (HR) measurement and the irregular rhythm notification (IRN) feature in stroke patients in sinus rhythm (SR) and AF. (2) Methods: Resting clinical HR measurements (every 5 min) were assessed using continuous bedside ECG monitoring (CEM) and the Fitbit Charge 5 (FC5). IRNs were gathered after at least 4 h of CEM. Lin’s concordance correlation coefficient (CCC), Bland-Altman analysis, and mean absolute percentage error (MAPE) were used for agreement and accuracy assessment. (3) Results: In all, 526 individual pairs of measurements were obtained from 70 stroke patients—age 79.4 years (SD ± 10.2), 63% females, BMI 26.3 (IQ 22.2–30.5), and NIHSS score 8 (IQR 1.5–20). The agreement between the FC5 and CEM was good (CCC 0.791) when evaluating paired HR measurements in SR. Meanwhile, the FC5 provided weak agreement (CCC 0.211) and low accuracy (MAPE 16.48%) when compared to CEM recordings in AF. Regarding the accuracy of the IRN feature, analysis found a low sensitivity (34%) and high specificity (100%) for detecting AF. (4) Conclusion: The FC5 was accurate at assessing the HR during SR, but the accuracy during AF was poor. In contrast, the IRN feature was acceptable for guiding decisions regarding AF screening in stroke patients.

## 1. Introduction

Cryptogenic stroke, with a prevalence of 30% of cases, is a subtype of ischemic stroke without an identifiable cause after a comprehensive evaluation. However, there are several potential underlying causes that must be studied to guide treatment and secondary prevention strategies. One of the most common causes is undetected atrial fibrillation (AF) [1].

In general, stroke is a major public health problem worldwide with a significant burden on healthcare systems in terms of costs and resources. The estimated total cost of stroke in Europe is over EUR 60 billion per year [2]. Unfortunately, the burden of stroke is expected to rise in the incoming years with a parallel increase in healthcare service demand. To address the challenge, new stroke care pathways are needed to improve the efficiency of stroke care.

It has been demonstrated that AF is one of the main causes of recurrent stroke, and people with AF are about five times more likely to have a stroke than those without AF [3]. Moreover, the recurrence of ischemic stroke in people with AF is also more likely to be severe and disabling [4]. Therefore, heart rhythm monitoring to detect AF remains the basis of the follow up of cryptogenic stroke survivors, since identifying this condition may optimize secondary prevention measures, such as anticoagulation therapy, to reduce the risk of recurrent ischemic stroke [5,6]. However, in some cases, AF is usually complex to detect using conventional cardiac monitoring due to its paroxysmal nature, which means intermittent AF episodes of irregular and rapid heartbeats that stop on their own without treatment and are not usually associated with noticeable symptoms [7]. Consequently, innovative methods to increase the rate of early AF detection in stroke patients using a simple and cost-effective screening protocol should be considered [8].

New technologies are being developed to increase the rate of AF detection using prolonged rhythm monitoring, including implantable cardiac monitors (ICMs) that can be implanted under the skin to monitor the heart rhythm and detect episodes of AF over a longer period of time. Unfortunately, ICMs carry some disadvantages, such as invasiveness and high upfront cost, particularly when compared to traditional monitoring methods such as external Holter monitoring [9]. However, it is important to note that cost is just one factor to consider when determining the appropriateness of a monitoring method. Other factors, such as patient preference, clinical judgment, availability of resources, and health insurance coverage may also play a role in decision making.

Commercially available smartwatches allow the collection of real-time heart rate (HR) data passively and continuously as well the detection of irregular heart rhythms through a non-invasive optical technology called photoplethysmography (PPG), which measures the pulse through a light sensor [10]. It has been reported that smartwatches may be a cost-effective method for guiding AF detection in high-risk populations because they offer some advantages when compared to ICMs, such as non-invasiveness, cost, convenience, extra features, and acceptance [11]. While PPG technology is less accurate than electrocardiogram (ECG) technology, studies have shown that it can be effective for detecting AF in some patients, particularly those with paroxysmal AF or with a high risk of developing AF [12]. Furthermore, it was published that a PPG-based algorithm with an input of an irregular heart rhythm demonstrated a 98% positive predictive value to detect AF [13].

Overall, the use of PPG-based smartwatches to increase the rate of AF detection in stroke patients holds promise for improving outcomes and reducing the risk of complications, such as recurrent stroke. However, further research is needed to evaluate the effectiveness of these methods in clinical practice. We conducted a pilot study to assess the HR measurement as well as the irregular rhythm notification feature of a consumer smartwatch to detect AF in stroke patients.

## 2. Methods

### 2.1. Participants

A cross-sectional pilot study was carried out in the Vall d’Hebron University Hospital’s Comprehensive Stroke Center from August 2022 to November 2022. This pilot study protocol was designed in accordance with the development of a clinical trial registered prospectively at clinicaltrials.gov (NCT05565781). The study met good clinical practice guideline standards, including the Declaration of Helsinki.

Consecutive patients who underwent continuous bedside ECG monitoring (CEM) to monitor HR in the stroke unit were included. The exclusion criteria were having a pacemaker, incapability to give informed consent, intravenous line at both forearms, and low-quality CEM signal. During the validation for the assessment of HR measurements, all patients were at rest and supine on a hospital bed with their arms resting on the bed’s surface.

### 2.2. Heart Rate Assessment

CEM was carried out using the Philips IntelliVue MX500^®^ to obtain the reference HR. This device is capable of calculating the time-series data of the HR from the ECG data. The measured HR values were collected manually by one trained member of the research staff from the device management dashboard of the CEM system. The time-series CEM-HR recordings were divided into 5-min segments (Figure 1), and each CEM-HR segment was classified as either non-AF or AF. Before collecting data, the quality of the signal of the CEM system was manually reviewed to ensure ECG measurement and HR detection.

The PPG-based smartwatch employed was the Fitbit Charge 5^®^, and the Fitbit app was installed on an Android smartphone to wirelessly connect to the smartwatch. The HR data are transmitted from the PPG-based smartwatch to the smartphone via a Bluetooth connection. From the smartphone, the raw PPG data are transferred to a cloud server and forwarded to the cloud-based Fitbit application for analysis. The resulting HR data are then displayed in the Fitbit dashboard (Figure 2).

Cumulative time-series data pertaining to HR measurements in real time were directly collected from the Fitbit website dashboard. There were no missing values regarding HR data from the PPG-based smartwatch. The HR measurement recordings by the Fitbit Charge 5^®^ are shown in Figure 3.

During data acquisition, the smartwatch was placed on the patient’s non-dominant wrist. The device was affixed securely above the ulnar styloid. In case of severe edema, rash, or damage in the non-dominant wrist, the smartwatch was placed on the dominant one. The wrist setting was configured before each validation in the Fitbit app to match the wrist the patient wore the device on (dominant or non-dominant). The Fitbit Charge 5^®^ was attached to the patient’s wrist as shown in Figure 4.

For each patient, the smartwatch measured the HR at rest, and simultaneously, CEM data were collected every 5 min. Resting HR recordings were collected after the patient was lying in a supine position for at least 30 min before the study validation. We extracted continuous segments during monitoring from each patient. Time synchronization of the HR data was performed based on the time stamps associated with the HR data acquired from each device.

Figure 5 displays the process of analyzing the smartwatch and CEM-HR signals. The processed PPG-HR signals from the smartwatch dashboard were divided into 5-min segments, similar to the CEM-HR data.

### 2.3. AF Detection Assessment

Since one of the aims of this study was to evaluate the accuracy obtained in AF detection by the Fitbit’s irregular rhythm notification (IRN) feature in comparison to that of AF detection by CEM, patients wore the Fitbit device and underwent CEM while resting on a bed, and the Fitbit app was set to alert if an irregular rhythm was detected. Thus, the data collected by the Fitbit app and the CEM were compared to identify instances of AF (Figure 6). To test the IRN feature, a minimum of 4 h of CEM was required, and each patient was classified into the non-AF rhythm or the AF rhythm group based on CEM signals. If the patient exhibited any AF events during the continuous 4-hour test, the patient was classified into the AF group; otherwise, the patient was classified into the non-AF group. AF events detected by CEM were validated by a blinded physician.

Once the estimated monitoring period was completed, the trained member of the research staff collected the irregular rhythm notifications by opening the Fitbit app on the Android smartphone (Figure 7).

The Fitbit’s irregular rhythm notification (IRN) feature uses a PPG-based algorithm to detect heart rhythm patterns that may indicate AF. The PPG data collected by the Fitbit device are sent to the servers, where they are processed using the algorithm to detect irregular rhythms. If an irregular rhythm is detected, a notification is sent back to the user’s device (Figure 2). The specifics of Fitbit’s software algorithm are confidential and not made available to the public. However, this PPG-based algorithm requires at least 11 consecutive, irregular tachograms (pulse windows) within a 5-min overlapping period to produce an IRN [13].

### 2.4. Data Analysis

Data were processed using STATA Software version 14. At first, the demographic variables (age, sex, BMI), history of AF episodes, and NIHSS score status at the time of the monitoring period were summarized using standard descriptive statistics. Data with normal distribution were presented as mean ± standard deviation (SD). Meanwhile, data with non-normal distribution were presented as medians (interquartile range (IQR)). Categorical variables were presented as numbers and percentages. Hence, measurement agreement and accuracy were evaluated through standard methods, as indicated in the following paragraphs.

### 2.5. Agreement Analysis

The agreement between HR measurements obtained from the smartwatch and the CEM was assessed using Lin’s concordance correlation coefficient (CCC), and these values were then graphically plotted according to Bland and Altman analysis. In particular, the CCC can provide information on the strength and direction of the agreement as well as on the extent to which the two methods differ in their means and standard deviations. The CCC can also be used to assess the relative importance of correlation and bias in the agreement [14].

On the other hand, Bland-Altman analysis is a graphical method for assessing the agreement between two methods of measurement or between a model’s predictions and the actual values. It plots the differences between the two methods or values against their means and calculates the limits of agreement, which represent the range within which 95% of the differences are expected to fall [14].

We calculated the CCC and the associated 95% confidence intervals to provide a measure of agreement for the paired observations from the smartwatch and the CEM. The CCC ranges between −1 and 1, where a value of 1 indicates perfect agreement, a value of 0 indicates no agreement, and a value of −1 indicates perfect disagreement. A CCC of >0.8 was considered to indicate good agreement in HR measurement [14]. 

### 2.6. Accuracy Analysis

Accuracy was defined based on the paired differences between each of the HR recordings, using the CEM-HR as the standard. The paired differences were calculated as (HR CEM − HR smartwatch). Then, the absolute values of the paired differences were calculated to better assess the magnitude of the difference irrespective of the direction. Finally, the mean absolute percentage error (MAPE) relative to the HR data was calculated by averaging the individual absolute percentage errors of the HR measurements. The formula for calculating the MAPE was as follows:MAPE = [(HR CEM − HR smartwatch)/HR CEM] × 100%

In this study, MAPE expressed the percentage difference of the HR acquired by the smartwatch relative to that of the CEM, and a value < 10% was considered reliable accuracy [15].

### 2.7. Accuracy of the Irregular Rhythm Notification Feature

Assessment of the diagnostic accuracy of the IRN feature involved evaluating its performance in terms of sensitivity, specificity, positive predictive value (PPV), and negative predictive value (NPV). The diagnostic accuracy of the IRN feature was tested against the reference monitor (CEM-AF reference) using the MedCalc Software [16]. The positive and negative predictive values (PPV and NPV) were also estimated on the basis of an expected AF prevalence of 15% among patients with ischemic stroke [17].

## 3. Results

During the development of the study, 76 stroke patients were screened, of whom 6 were excluded due to lost CEM signals. Then, 70 stroke patients successfully completed the HR assessment, of which 33 completed a round of 4 h of simultaneous ECG monitoring by CEM and AF data collection (IRN feature) (Figure 8). The mean age was 79.4 years (SD ± 10.2); 62.9% were female; the median BMI was 26.3 (IQ 22.2–30.5); and the median NIHSS score was 8 (IQR 1.5–20). A history of AF diagnosis was present in 67.1% (47/70) of patients. Table 1 displays the baseline characteristics of the patients.

A total of 526 individual pairs of HR measurements were collected from the Fitbit Charge 5 to determine the agreement of the HR measured by this device with that obtained by the reference method (CEM). For this, patient’s HR recordings were classified as sinus rhythm or AF during the measurements (Table 2).

Figure 9 shows Lin’s concordance correlation coefficient (CCC) on paired HR data in scatterplots. For sinus rhythm, the CCC of the Fitbit Charge 5 was 0.791 (95% CI 0.350–0.452; Figure 9a). However, the CCC of the Fitbit Charge 5 during AF episodes was 0.211 (95% CI 0.148–0.273; Figure 9b).

The overall bias of the HR measurements by the Fitbit device based on the reference values (CEM-HR) was determined using the Bland-Altman plot. In general, this analysis revealed that the Fitbit Charge 5 tended to underestimate HR values with a mean difference of 10.18 bpm (Figure 10). The limits of agreement (LoA) were quite broad, indicating that the measurements were less accurate over 90 bpm. These results confirm the tendency already observed in the scatterplots. The Bland-Altman plot showing the mean differences and limits of agreement is shown in Figure 10.

Regarding the accuracy metrics of the HR measurements, we calculated the MAPE between all paired measurements of the Fitbit device and the reference values (CEM-HR). In sinus rhythm, the smartwatch had a MAPE of 6.18% (95% CI 4.12–8.24%). During an AF episode, the smartwatch had a MAPE of 16.48% (95% CI 11.64–21.67%) (Table 3). Figure 11 shows a graphical representation for summarizing and visualizing these statistical data.

We also searched for the influence of HR ≥ 100 bpm in the measurement accuracy of the Fitbit device. Table 4 shows the results of our analysis, and the respective box plot is presented in Figure 12. As expected, dividing patients according to HRs above 100 bpm led to the observation of a significant difference (*p* < 0.05) in the sinus rhythm group and the AF group. We identified that HR ≥ 100 bpm significantly worsened the accuracy of the Fitbit device’s HR measurements during an AF episode (Figure 12).

The IRN feature was collected from a total of 33 patients (Figure 7). Participants underwent 4 h of simultaneous ECG monitoring by CEM and continuous PPG-HR recording using a Fitbit smartwatch. All the AF events referenced by CEM were recognized as AF by a blinded physician. A total of 29 (87.8%) of patients had AF during the 4-hour monitoring period. The IRN results matched the occurrence or absence of an AF episode based on the CEM system in 42.4% (14/33) of cases. Accordingly, the IRN feature had a sensitivity of 34.5% (95% CI 17.9–54.3%), and a specificity of 100% (95% CI 39.8–100%) for the detection of AF when compared to CEM. On the basis of an expected prevalence of 15% in the population, a PPV of 100% and an NPV of 86.6% (95% CI 86.9–91.9%) were estimated.

## 4. Discussion

In this study, we aimed to investigate the ability of a commercially available PPG-based smartwatch to aid physicians in AF screening and remote HR monitoring in stroke patients. To our knowledge, this is the first study to evaluate the accuracy of real-time average HR readings from PPG-based smartwatches reported in their health metrics dashboard. This study is part of a pilot study to establish the feasibility of an AF screening strategy for stroke patients using a consumer smartwatch (ClinicalTrials.gov Identifier: NCT05565781).

There were two different approaches tested: (1) to estimate the agreement between resting clinic HR measurements obtained by the smartwatch and by CEM, and (2) to assess the feasibility of an IRN feature that helps identify signs of AF in stroke patients.

We observed that the average HR collected from the dashboard of this PPG-based smartwatch tended to underestimate the real resting HR and that the bias was larger as the HR increased above 90 bpm. Regarding resting HR dynamics, we found that smartwatch-HR demonstrated a high concordance with CEM-HR in sinus rhythm, but a relatively weak correlation was observed during AF episodes (Figure 9). Consistently, the MAPE for resting HR was >10% during AF and lower during sinus rhythm.

PPG-based smartwatches are specially designed to meet fitness goals and monitor HR during exercise. However, resting HR measurement via a smartwatch has not been properly evaluated for clinical purposes. Previous studies have suggested that PPG-based smartwatches underestimate the HR measurements when compared to a reference method or standard depending on the specific device and the conditions in which it is being used, such as physical activity level or clinical conditions [5,10].

Many studies reported that the accuracy of PPG-based smartwatches for HR measurement during physical activity is generally lower than that during rest [10,12,15,18]. For example, a recent study published in 2022 by Alfonso et al. involving PPG-based, wearable devices reported that this technology demonstrated good-to-excellent agreement in measuring HR during rest. However, the authors noted that the devices tended to underestimate HR during high-intensity exercise compared to a reference device. Overall, the study suggests that PPG-based HR sensors in smartwatches can provide fairly accurate readings during rest [18]. Nevertheless, these studies focused mainly on healthy and younger populations, and it is important to consider the potential differences between younger and elderly populations when interpreting the results.

On the other hand, when referring to clinical conditions, several studies have compared the accuracy of PPG-based HR sensors in smartwatches to electrocardiogram (ECG) measurements during AF episodes, and the results have been mixed. Some studies have reported good agreement between PPG-based HR measurements and ECG measurements, while others have found significant differences between the two methods [19].

Since we found significant differences for the smartwatch during AF with fast heartbeat, studies focused on the accuracy of HR estimation from smartwatches in tachyarrhythmias at rest were reviewed. For example, a study published by Koshy et al. in 2018 evaluated the accuracy of smartwatch HR measurements during rest in hospitalized patients, most of them aged 60 years and older, and found that several popular devices, including the Fitbit, had lower accuracy compared to a medical-grade ECG for HR in AF [20]. Another study published by Sequeira et al. in 2020 to evaluate the accuracy of different wearable devices in measuring HR during supraventricular tachycardia (SVT) found that the Fitbit Charge 2 had lower accuracy in measuring HR during SVT compared to other devices among middle-aged and older patients with documented reentrant SVT [21]. In 2020, Al-Kaisey et al. also reported that HR during AF was underestimated compared to that during sinus rhythm by wrist-worn heart rate monitors, including the Fitbit Charge 3. This study found that the Fitbit Charge 3 had lower sensitivity (75%) and positive predictive value (74%) while assessing HR monitoring during AF compared to a 12-lead electrocardiogram (ECG) among ambulant patients aged 55 and over [22].

Our findings may be driven by the physical characteristics of our study population, especially those related to age, considering that the mean age of our patients was around 79 years. PPG technology estimates the HR by detecting subtle variations in the color of the skin, which correspond to changes in blood volume in the microvascular bed of tissue beneath the skin caused by the contraction and relaxation of the heart; however, compared to younger people, elderly people may have changes in the blood volume in their cutaneous microcirculation that can affect PPG measurements of HR [23]. As people age, their skin becomes thinner and less elastic, which can cause a decrease in blood flow to the microvascular bed. This reduction in blood flow can make it more difficult for PPG sensors to detect changes in blood volume and accurately estimate HR. Additionally, elderly people may have other factors that can affect the accuracy of PPG measurements, such as medication use, chronic health conditions, or reduced physical activity levels. These factors can affect blood circulation and lead to changes in the microvascular bed, which can in turn affect PPG measurements [23].

Overall, while PPG technology is generally effective at estimating HR in people of all ages, there may be some age-related differences in the accuracy of the measurements due to changes in the cutaneous microcirculation and other factors. This poses a challenge as to how smartwatches could help achieve preventive goals in elderly stroke patients. Further studies to assess the HR are needed in order to recommend whether the smartwatch is a reliable tool for monitoring HR in this population.

In general, studies have shown that PPG-based HR sensors in smartwatches can provide reliable HR measurements during sinus rhythm but may be less accurate during AF episodes. The accuracy of HR measurements during AF episodes can be affected by various factors, such as the irregularity of the heartbeat, motion artifacts, and changes in blood flow. Overall, while PPG-based HR sensors in smartwatches can be a useful tool for monitoring the HR, they may not be as accurate as ECG measurements during AF episodes, especially in challenging situations, such as with elderly stroke patients. It is important to use caution when interpreting HR measurements from smartwatches during AF episodes. Therefore, smartwatches may not be recommended to guide therapeutic decisions to control the HR in patients with AF. Researchers are working on the design of new PPG-based algorithms to provide better detection accuracy for the HR in various rhythm scenarios including AF [24].

In contrast, promising results were reported regarding the IRN feature available on smartwatches. The IRN feature in smartwatches, such as the Fitbit devices, works by using PPG sensors, which continuously monitor the wearer’s heart rate and heart rhythm. If the device detects an irregular rhythm that is consistent with atrial fibrillation or other arrhythmias, it will alert the wearer with a notification. Recently, the Fitbit Heart Study, a large-scale clinical trial, was conducted by Fitbit in collaboration with the American Heart Association (AHA) and other organizations to evaluate the accuracy of the IRN feature on the Fitbit devices for detecting AF. The study involved over 400,000 participants who wore a Fitbit device that was equipped with the IRN feature for a period of up to one year. The results of the study, which were published in the Journal of the American College of Cardiology in 2022, showed that the IRN feature had a sensitivity of 84% and a specificity of 93% for detecting AF compared to ECG measurements. The authors of the study concluded that the IRN feature on Fitbit devices may be a useful tool for detecting AF and other arrhythmias in individuals at risk [13].

Considering our results regarding the low sensitivity of the IRN feature, two in three cases of AF could be undetected by the PPG-based algorithm. From the perspective of screening programs for AF, where the sensitivity must be high, the use of the IRN feature should be complemented with more data, such as the age of the patients, biomarkers, and chronical conditions, to improve the accuracy of the algorithm. Conversely, the specificity of the IRN feature for the group of patients studied seems to provide very good reliability as all cases of AF were confirmed. In this way, ruling out false-positive alarms of AF may reduce patient anxiety and cardiologist overload. Nevertheless, the risk of misdiagnosis by the IRN must be assessed in prospective studies.

We assumed that a short episode of AF was barely detected by the algorithm. Consequently, the PPG-based algorithm may be more effective in patients with prolonged AF episodes. It has been reported that the IRN feature warned of an AF episode at least a few hours after it occurred, so it is not a real-time tool to detect an active AF episode unless the episode endures for a long time [19]. Therefore, it is important to consider the overall accuracy and limitations of the PPG-based irregular rhythm algorithm feature in the context of early detection and management of AF in specific patient populations, such as older stroke patients, and to use it as an adjunct to standard diagnostic methods to improve AF diagnostics. This tool may be use as part of a low-cost screening strategy to accurately guide most AF diagnosis interventions by improving patient selection [25].

Overall, smartwatches have the potential to improve the management of stroke patients in the long term. In patients with cryptogenic stroke, after standard-of-care cardiac monitoring, wearing a smartwatch could uncover episodes of AF by means of the IRN feature. However, it is important to note that smartwatches should not replace regular medical care and should be used in conjunction with medical assistance.

When interpreting our findings, a few limitations must be considered when talking about the IRN feature’s accuracy. In our study, we did not consider the inclusion of patients who presented with other arrhythmias different from AF. It would be important to know the performance of this PPG-based algorithm against other types of arrhythmias. For use as part of an AF screening program, the high specificity of this method can be of great advantage to identify undiagnosed patients with high risk. However, the nature of this study’s design did not enable us to investigate this further. Most notably, our study was designed to include only stroke patients in hospital settings. Our findings and their implications should not be translated to ambulatory environments. In addition, the high median age of the patients may have affected the results regarding the PPG assessment. With regard to the accuracy of the HR measurements, the impact of other clinical variables, such as stroke symptoms, was not considered in this report. Nevertheless, no PPG signals were discarded due to limb weakness.

## 5. Conclusions

Compared with our reference device, the smartwatch was accurate at assessing the HR during sinus rhythm. In contrast, data pertaining to HR measurements during AF showed poor accuracy. These findings suggest certain limitations when considering the application of real time-processed resting HR data from smartwatches in stroke patients within hospital settings.

The irregular rhythm notification feature appears acceptable for guiding decisions regarding AF screening through a better selection of stroke patients at higher risk of AF. This feature may facilitate other medical tests that can be conducted to enable earlier diagnosis of AF, providing stroke patients with a signal to seek medical attention and to potentially identify undiagnosed AF. Overall, more research is needed to determine its accuracy and effectiveness in rea-world settings using well-structured health interventions. It is important to note that the irregular rhythm detection features in smartwatches are not intended to replace clinical diagnosis and monitoring by a healthcare professional.

## Figures and Tables

**Figure 1 sensors-23-04632-f001:**
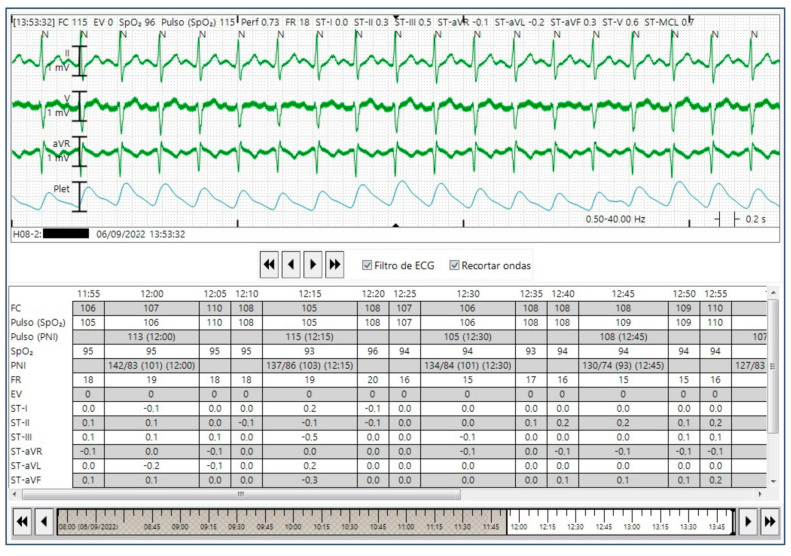
HR data from continuous bedside ECG monitoring (CEM) device.

**Figure 2 sensors-23-04632-f002:**
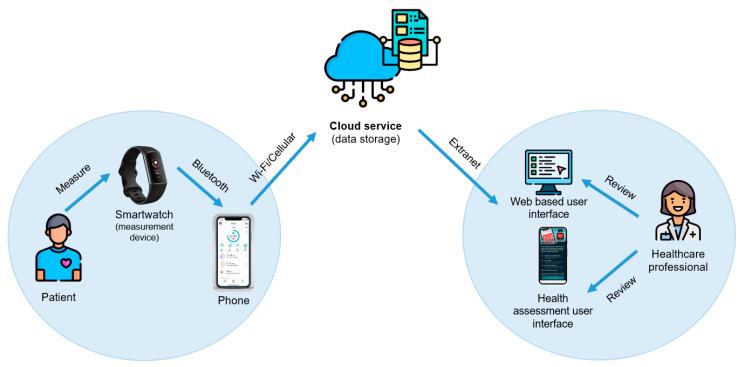
Smartwatch’s cloud data processing and storage.

**Figure 3 sensors-23-04632-f003:**
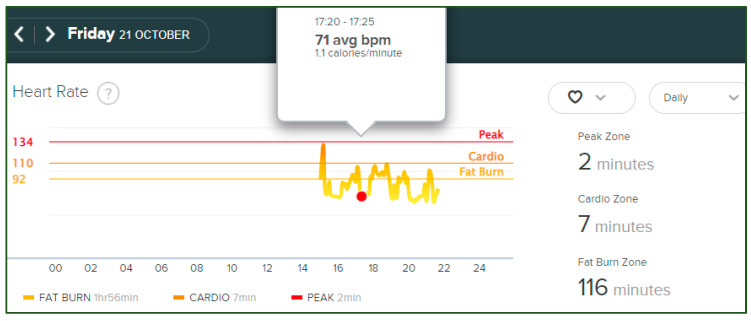
HR data from Fitbit Charge 5^®^ website dashboard.

**Figure 4 sensors-23-04632-f004:**
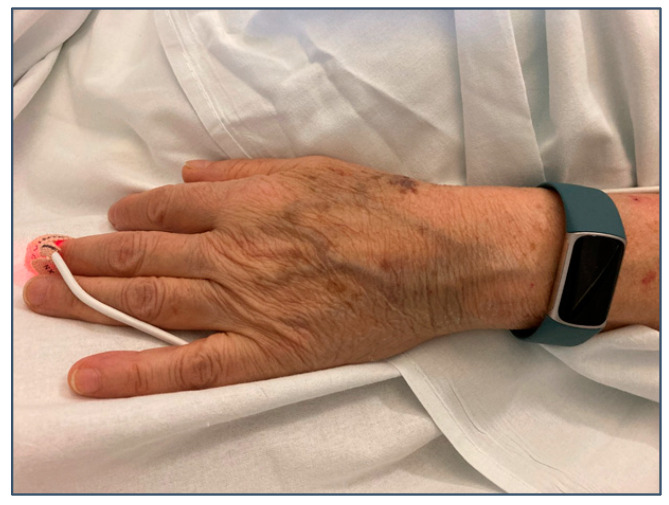
Fitbit Charge 5^®^ location.

**Figure 5 sensors-23-04632-f005:**
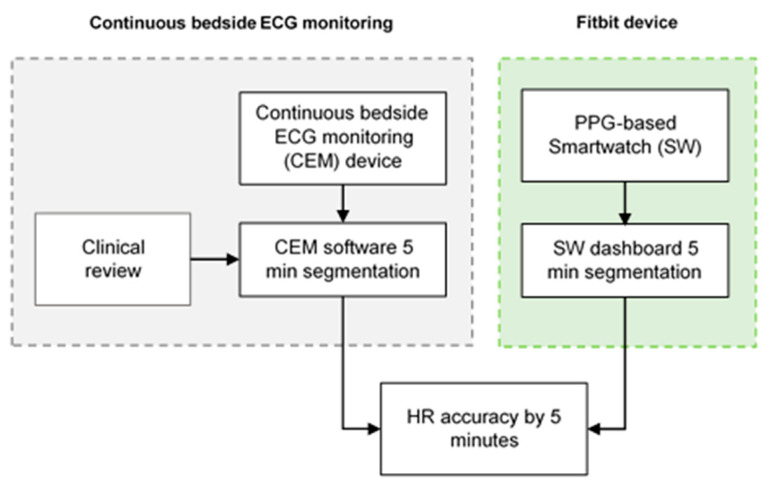
Data collection processes of HR recordings using continuous bedside ECG monitoring and PPG-based HR recordings from a smartwatch.

**Figure 6 sensors-23-04632-f006:**
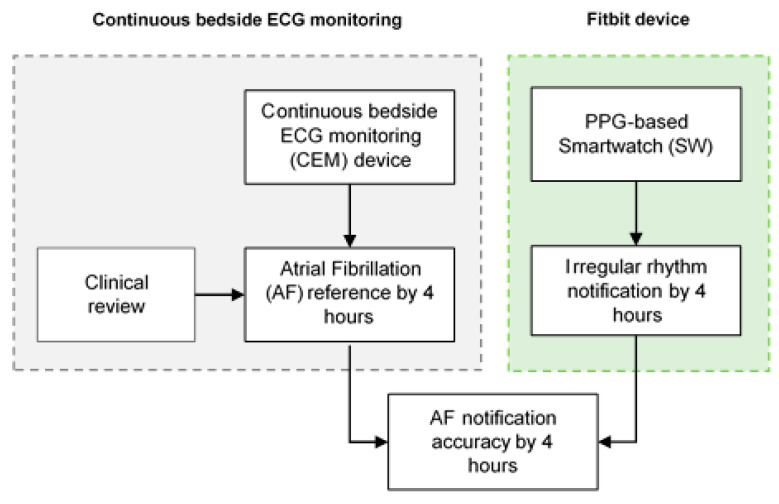
Data collection processes of irregular rhythm notification using continuous bedside ECG monitoring and PPG-based algorithm for detecting AF from a smartwatch.

**Figure 7 sensors-23-04632-f007:**
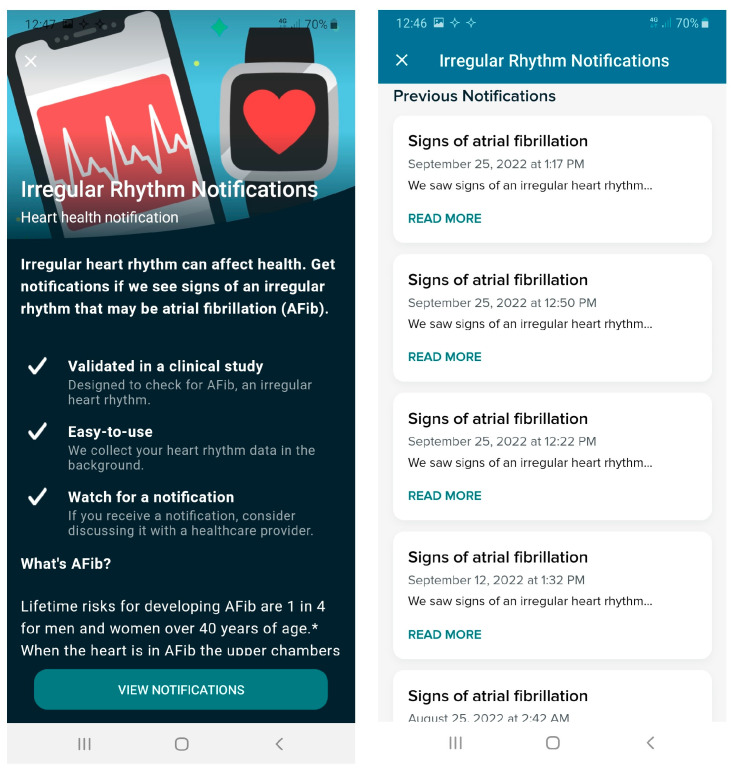
Assessment health data in Fitbit app showing irregular rhythm notifications.

**Figure 8 sensors-23-04632-f008:**
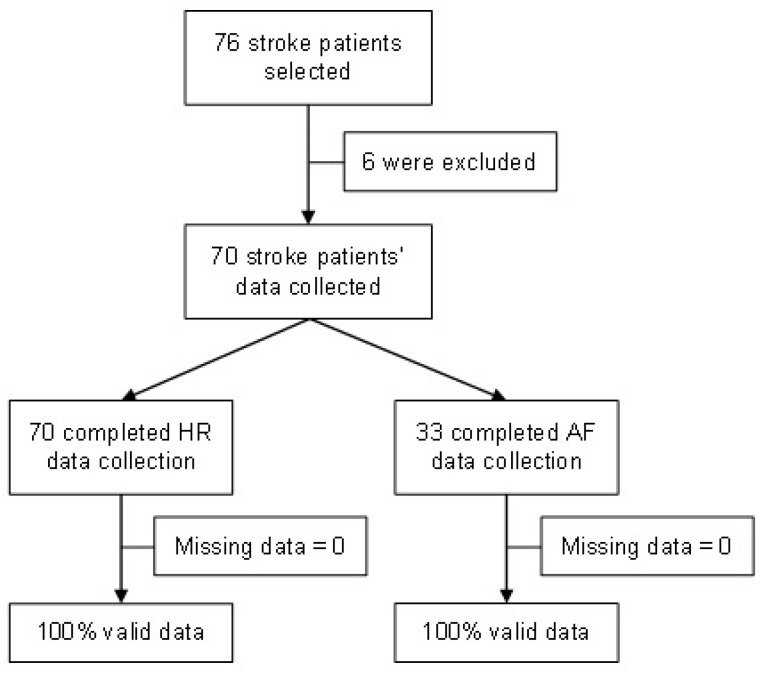
Participant selection and summary of results.

**Figure 9 sensors-23-04632-f009:**
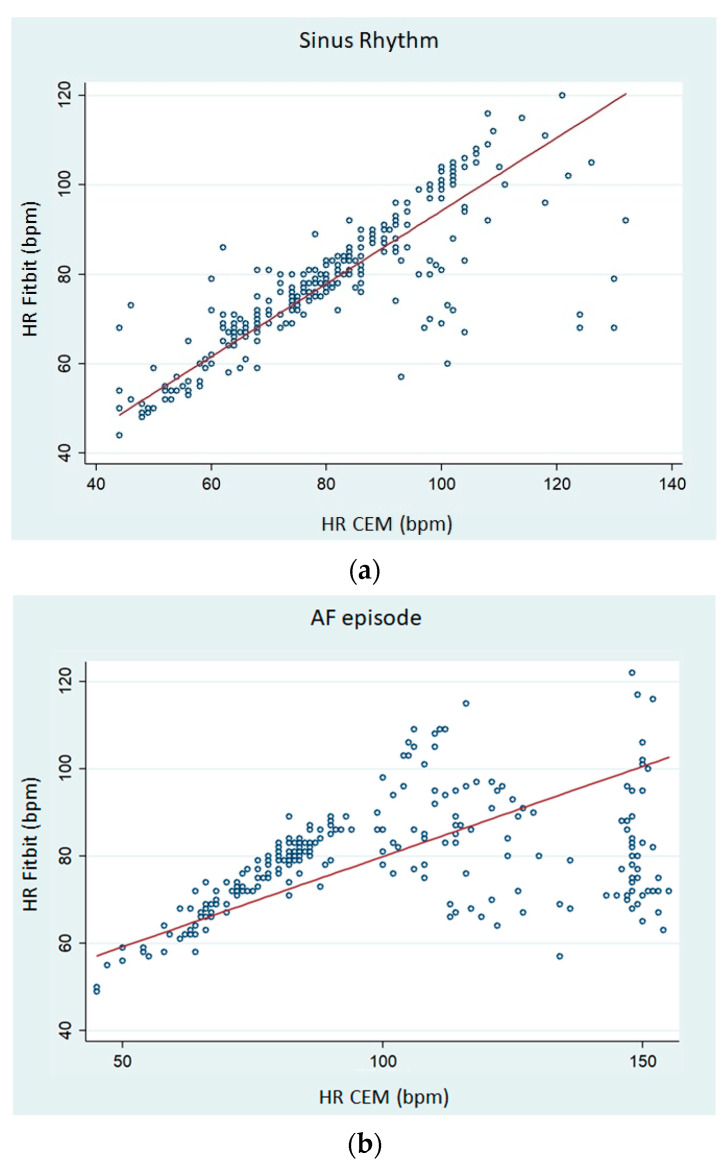
Scatterplots of pairs of HR measurements. (**a**) HR readings obtained from CEM and Fitbit Charge 5 in sinus rhythm. (**b**) HR readings obtained from CEM and Fitbit Charge 5 when AF is present.

**Figure 10 sensors-23-04632-f010:**
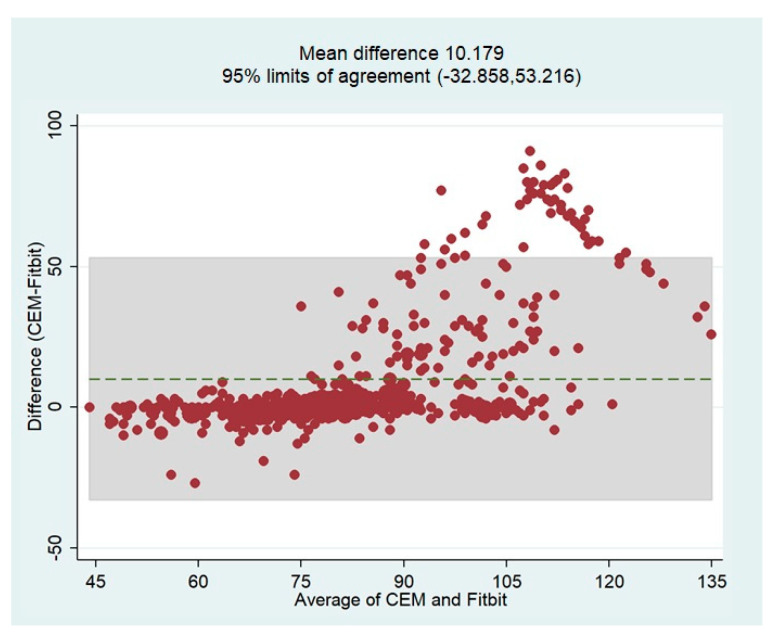
Bland-Altman plot of 5-min HR readings from the CEM and Fitbit Charge 5. The green, dashed line represents the mean difference between the tested device and CEM estimates (in 5 min). The gray-shaded area represents the 95% confidence interval around the mean differences ±1.96 SDs. Dots outside the gray-shaded area correspond to extreme error values.

**Figure 11 sensors-23-04632-f011:**
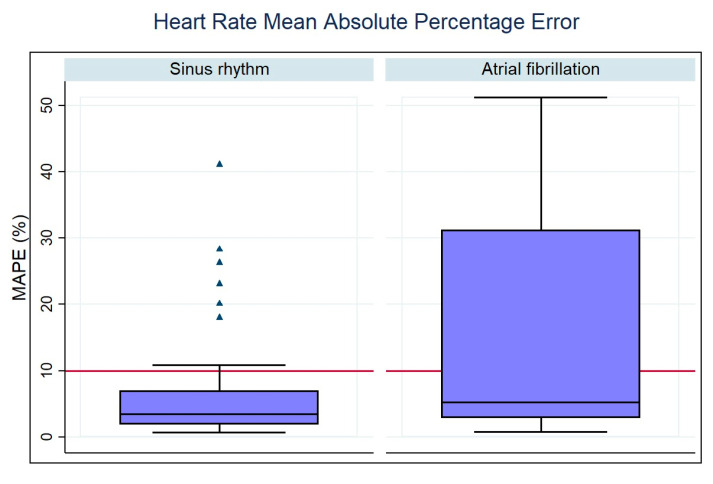
Box plot for the distribution of MAPE values according to AF or non-AF.

**Figure 12 sensors-23-04632-f012:**
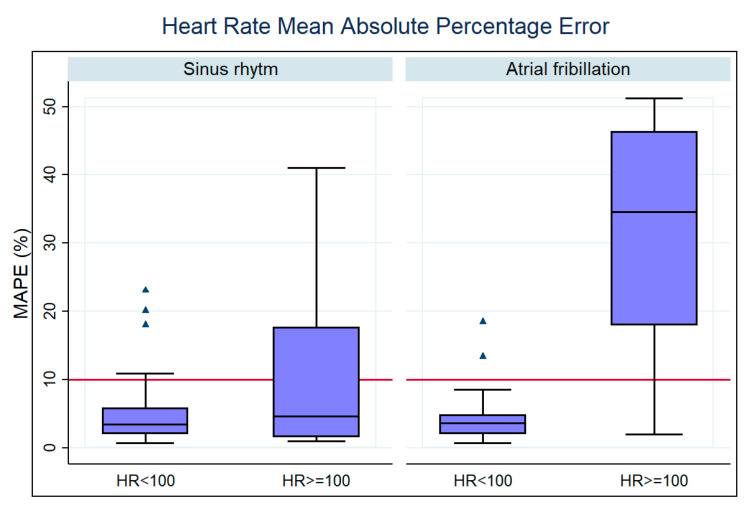
Box plot for the distribution of MAPE values according to AF or non-AF and HR ≥ 100 bpm or not.

**Table 1 sensors-23-04632-t001:** Characteristics of the patients (N = 70 patients).

Characteristic	Values ^1^
Age (years)	79.4 ± 10.1
Sex (female)	44 (62.9)
BMI (kg/m^2^)	26.9 ± 5.8
NIHSS score	8 (IQR 1.5–20)
AF history	47 (67.1)

^1^ Values are provided as n (%) or mean ± SD, unless specified otherwise. BMI: body mass index, NIHSS score: National Institutes of Health Stroke Scale, AF: atrial fibrillation.

**Table 2 sensors-23-04632-t002:** Lin’s concordance correlation coefficient (CCC).

SR or AF Presentation	CCC 95% CI
HR measurements in SR	0.791 (0.750–0.832) *
HR measurements in AF episode	0.211 (0.148–0.273) *

* Correlation is significant (*p* < 0.001). SR: sinus rhythm, AF: atrial fibrillation, CI: confidence interval.

**Table 3 sensors-23-04632-t003:** Mean absolute percentage error (MAPE).

SR or AF	MAPE (%) ^1^	95% CI	*p*-Value
HR measurements in SR	6.18 ± 9.57	4.12–8.24	0.001
HR measurements in AF episode	16.48 ± 17.89	11.6–21.67	

^1^ Values are provided as mean ± SD. SR: sinus rhythm, AF: atrial fibrillation.

**Table 4 sensors-23-04632-t004:** Mean absolute percentage error (MAPE) according HR ≥ 100 bpm or not.

HR Measurements	MAPE (%) ^1^	95% IC	*p*-Value
During sinus rhythm			
HR < 100 bpm	4.98 ± 4.89	3.50–6.45	0.021
HR ≥ 100 bpm	10.72 ± 13.48	2.15–19.28	
During AF episode			
HR < 100 bpm	4.40 ± 3.84	2.85–5.96	0.000
HR ≥ 100 bpm	30.50 ± 16.45	23.39–37.61	

^1^ Values are provided as mean ± SD.

## Data Availability

Data are unavailable due to the sensitive nature of patient information and because public release was not part of informed consent.

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
