# Peer review of "Accuracy of a Smartwatch to Assess Heart Rate Monitoring and Atrial Fibrillation in Stroke Patients"

_sensors, 2023, doi:10.3390/s23104632_

Round 1
Reviewer 1 Report
This paper reports on a single centre cross-sectional pilot study to validate resting heart rate (HR) measurement and irregular rhythm notification (IRN) feature in stroke patients in sinus rhythm (SR) and atrial fibrillation (AF).
Abstract - the CCC of Fitbit Charge 5 in sinus rhythm of 0.791 (95% CI 0.350-0.452) can be added to the abstract. The concluding sentence lines 22-23 must distinguish the data for HR from the data for AF – same comment on line 396 in the text
Line 77 – spell CEM at first used in the main paper
Line 82 – 2.2 - I think this segment not only assesses heat rate but also AF? (see lines 89-90). If not, where is the segment that describes the methods for AF detection? Is it from line 132? If so, please place an appropriate sub-heading in the relevant location
Line 134 – while Fitbit can ‘detect irregular heart rhythms including AF”, not all irregular rhythm are due to AF. Thus line 135 ‘look for signs of AF’ may not be correct
Note line 128 where AF by CEM was validated by a blinded physician. For Fitbit, how was AF validated? More than simply irregular rhythm? Was the ‘irregular rhythm’ validated by a blinded physician?
To be consistent with how HR assessment was described (CEM then Fibit), perhaps the paras on AF can be similarly sequenced ie lines141-145(CEM) to precede lines 132-138 (Fitbit)
Fig 5 – if this is about heart rate, should AF appear here? AF seems appropriate as it is in Fig 6.
Lines 269 to 272 – please recheck the stats that with a sensitivity of only 34.5%, the PPV was 100%... My concern remains that not all irregular rhythm on Fitbit is AF on CEM
Line 293 – please reference the ‘previous studies’
Line 297 – please reference the ’several studies’ - i only see the one by Alfonso et al discussed
Line 396 - the concluding sentence must distinguish the data for HR from the data for AF – same comment on the abstract lines 22-23
Author Response
Dear Reviewer,
We really appreciate the time and effort to review our article. You will find the answer to each query in the attachment file. Every edition in the manuscript is in bold. Please let us know if any point precise clarification.
Please see the attachment file.
Best regards,
Ms. Claudia Meza on behalf of study group.

Reviewer 2 Report
- The study addresses an important research question regarding the use of smartwatches for AF screening in older stroke patients, but the introduction could benefit from a more explicit background and rationale. The methods are clearly stated, but more detailed explanations of data collection and participant criteria could improve the reader's understanding of the study. The results are clearly stated, but the authors could more effectively communicate the implications of the low sensitivity and high specificity of the IRN feature for detecting AF in older stroke patients. The conclusion could be more specific and focused on the key takeaways of the study, such as the limitations of using the Fitbit Charge 5 for detecting AF and the potential benefits and limitations of using the IRN feature for AF screening in stroke patients. Additionally, the authors could provide more guidance for future research directions.
Line 70: "d Methods" should be "2. Methods"
Overall, the article provides a thorough explanation of cryptogenic stroke and its association with atrial fibrillation (AF). The discussion of the limitations of current monitoring methods and the potential advantages of new technologies, such as smartwatches, is well-presented. The study design and methodology are clearly described, and the results are presented in an organized and understandable manner. However, there are a few areas where the article could be improved:
-
The introduction could benefit from a more detailed explanation of what cryptogenic stroke is and its prevalence. Additionally, it would be helpful to include information about the burden of stroke on healthcare systems and the potential cost savings associated with identifying underlying causes of stroke.
-
The discussion of smartwatches could be strengthened by including more information about the limitations of PPG technology and the potential challenges associated with integrating smartwatch data into clinical decision-making.
-
The article would benefit from a more detailed discussion of the implications of the study results for clinical practice, particularly with respect to how smartwatches could be used to guide management decisions for stroke patients.
-
The authors should consider including a discussion of the potential limitations of the study design and how these limitations may impact the generalizability of the results.
Overall, this is a well-written article with important implications for stroke management. With some minor revisions, it could be a valuable addition to the literature on this topic.
Author Response
Dear reviewer,
We really appreciate the time and effort to review our article. You will find the answer to each query in the attachment file. Every edition in the manuscript is in bold. Please let us know if any point precise clarification.
Please see the attachment file.
Best regards,
Ms. Claudia Meza on behalf of study group

Round 2
Reviewer 1 Report
The authors have addressed my concerns